# Growth and Physiology of Two Psammophytes to Precipitation Manipulation in Horqin Sandy Land, Eastern China

**DOI:** 10.3390/plants8070244

**Published:** 2019-07-23

**Authors:** Juanli Chen, Xueyong Zhao, Xinping Liu, Yaqiu Zhang, Yayong Luo, Yongqing Luo, Zhaoquan He, Rui Zhang

**Affiliations:** 1Northwest Institute of Eco-Environment and Resources, Chinese Academy of Sciences, Lanzhou 730000, China; 2School of Life Science, University of Chinese Academy of Sciences, Beijing 100049, China; 3Naiman Desertification Research Station, Northwest Institute of Eco-Environment and Resources, Chinese Academy of Sciences, Lanzhou 730000, China; 4Horticultural Technology Department, Hanzhong Agricultural Technology Extension Center, Hanzhong 723000, China

**Keywords:** Horqin Sandy Land, precipitation addition, precipitation reduction, growth, physiological response

## Abstract

The availability of water is the critical factor driving plant growth, physiological responses, population and community succession in arid and semiarid regions, thus a precipitation addition-reduction platform with five experimental treatments, was established to explore the growth and physiology of two psammophytes (also known as psammophiles) to precipitation manipulation in Horqin Sandy Land. Changes in coverage and density were measured, and antioxidant enzymes and osmoregulatory substances in both of the studied species were determined. Investigation results showed that the average vegetation coverage increased with an increasing precipitation, and reached a maximum in July. Under the −60% precipitation treatment, *Tribulus terrestris* accounted for a large proportion of the area, but *Bassia dasyphylla* was the dominant species in the +60% treatment. *T. terrestris* was found to have higher a drought stress resistance than *B. dasyphylla.* From days 4 to 7 after rainfall, *B. dasyphylla* under precipitation reduction showed obvious water stress. The malondialdehyde (MDA) content of *B. dasyphylla* was higher than that of *T. terrestris*, but that of *B. dasyphylla* had the lower relative water content (RWC). The MDA content in the precipitation reduction treatments of the two studied species was higher than that in the precipitation addition treatments from days 4 to 10. Peroxidase (POD) and superoxide dismutase (SOD) activity and the soluble proteins and free proline content of *T. terrestris* were higher than those of *B. dasyphylla*. The free proline content of *T. terrestris* and *B. dasyphylla* increased with increasing drought stress. Our data illustrated that *T. terrestris* had a higher drought stress resistance than *B. dasyphylla*, which was correlated with the augmentation of some antioxidant enzymes and osmoregulatory substance. The adaptive mechanism provides solid physiological support for an understanding of psammophyte adaptation to drought stress, and of community succession or species manipulation for desertified land restoration.

## 1. Introduction

Water is a key driving factor in arid and semi-arid sand ecosystems. Plant growth and physiological processes are closely correlated with water availability. Precipitation is the most important source of water in arid and semi-arid sand areas, and is also a main constraint on the formation and development of a psammophyte community [1]. With the intensification of global climate change, rainfall events characterized by long intervals and high single rainfall amounts will increase. It is widely accepted that climate variation has caused changes in vegetation distribution patterns and productivity [2]. Furthermore, plant physiological features, such as photosynthesis and transpiration, antioxidant enzymes and osmoregulatory substances, are affected by climate variation [3,4]. As such, the responses of the above physiological features influence plant community dynamics, such as degradation, restoration and succession.

Horqin Sandy Land is located in the transitional zone of semi-arid and semi-humid regions in eastern Inner Mongolia of China [5]. The annual precipitation is approximately 351.7 mm, but this has shown a decreasing trend in recent years [6]. The effect of precipitation upon vegetation in this area is mainly reflected in the change of annual herbaceous plant species [7]. The disappearance and invasion of annual herbaceous plants occurs frequently with changes in precipitation in fixed dunes [8].

For grasslands in semi-arid regions, the rainfall timing pattern and the number of single rainfall events during the growing season have a greater impact on the net primary productivity than the total rainfall [9]. The Memorial model, as an estimate index of grassland potential productivity, points out that precipitation is one of the rigid limiting factors for the improvement of grassland in eastern Inner Mongolia [10]. In different months of the growing season, precipitation had different effects on grassland vegetation growth, especially from June to August (summer) [11]. In addition, the species diversity index varied with precipitation change, and the impact of this growing season precipitation on species diversity was greater than the impact of the total annual precipitation [10]. Studies revealed that vegetation coverage in most parts of inland China increased significantly. Precipitation in Inner Mongolia showed a bipolar change [12,13], with the western part showing a trend of warming and humidification, and the central and eastern parts characterized by warming and drying. Therefore, precipitation change inevitably affected not only plant community composition, diversity and productivity on a small scale, but also the structure and functioning of the entire ecosystem on a regional or even a global scale [14].

A study of the physiological responses of psammophytes to adverse stresses, which were mainly concentrated on sand burial, cold, drought and other stresses [15], indicated that antioxidant enzymes and osmoregulatory substances played an important role in resisting or repairing any damage caused by stress. However, when stress exceeded the tolerance limits of the plants, it resulted in plant growth stagnation or even death [16]. Under conditions of soil drought, rehydration, frequent dehydration and rehydration cycles, the responses of psammophytes are demonstrated, and many studies are manually simulated with a short-term regulation of soil moisture. It is difficult to systematically reflect the true responses of plants in their natural state. Studies on the physiological responses of *Digitaria sanguinalis* and *Setaria viridis* under simulated drought have been carried out [17,18]. The results show that osmoregulatory substances increase under drought stress, and moderate drought induces an increase in antioxidant enzyme activity, but severe drought damages the antioxidant enzyme system.

*Tribulus terrestris* is distributed in sandy land, uncultivated land, hillsides and residential areas. There are also many distributions in agricultural and forestry production land in the protected areas and the interlaced zone between the oasis edge and the agricultural area. Its stems lie flat and up to about 1m in length [19]. *B. dasyphylla* is a pioneer plant in semi-fixed or fixed dunes, flat sandy land and moderately saline-alkali land. Its stem is erect and its height is about 30–50 cm. It is often scattered or clustered in the grasslands, semi-deserts and desert areas of northern China [20,21]. Both plants are annual herbs [19,20,21], but their coverage and density varies greatly under different precipitation, thus it is necessary to reveal the differences in their physiological levels.

Studies on the physiological resistance of psammophytes to stress have mostly focused on short-term drought stress [22], which makes it difficult to precisely trace the physiological adaptation processes of the different plants of different families and genera in a short processing time and under a high intensity of treatments. Thus, the objectives of this study were as follows: (1) To reveal the growth and physiological responses of two psammophytes to precipitation gradients, including extreme drought under natural conditions, and (2) to examine the adaptive mechanisms of the two species to a sand habitat.

## 2. Materials and Methods

### 2.1. Site Description

This study was carried out in the Naiman Desertification Research Station (42°58′N, 120°43′E), Chinese Academy of Sciences, which is located in the south-eastern part of Horqin Sandy Land, eastern Inner Mongolia, China. This area is characteristic of a continental semi-arid monsoon climate, with a mean annual precipitation of 351.7 mm, 70–80% of which falls from June to August (summer). The annual mean potential evaporation is 1935 mm and the annual frost-free period is approximately 150 days. The landscape of this region is characterized by sand areas and lowland areas. Most of the lowlands have been reclaimed into cropland, while the sand areas are used as pasture. The sandy soil mainly consists of coarse sand and silt. The natural vegetation mainly consists of *Agriophyllum squarrosum, Setaria viridis, Corispermum marocarpum, Salsola collina, Artemisia halodendron, Caragana microphylla, Chenopodium glaucum, Bassia dasyphylla* and *Tribulus terrestris* [23,24].

### 2.2. Experimental Design

A precipitation addition-reduction structure was constructed in April 2011 and used in this study from 10 May to 20 August 2016. The device was 2 m (length) × 2 m (width) × 1.5 m (height). The structure had a square steel frame, with the top tilted by 15 degrees to support rain intercept grooves for rain reduction, and a rain tank for rain collection (Figure 1). The rain intercept groove was made of highly transparent 5 mm-thick Perspex with a light transmission of 95%. The rain tank was made of a stainless-steel plate with a thickness of 3 mm and sealed ends. White aluminum-plastic pipes were connected to the bottom of the rain tank for rain addition, and nine isometric holes were made evenly along each of the pipes for the redistribution of the collected rain [25].

According to the annual precipitation data for the last 49 years in this area, the maximum precipitation reached 567.1 mm in 1986, which was 61.2% over the average, and the minimum precipitation was 213.1 mm in 2000, 39.4% less than the average precipitation. Five treatments were set up: (a) Control, without precipitation addition or reduction; (b) precipitation added by 30% (+30%); (c) precipitation added by 60% (+60%); (d) precipitation decreased by 30% (−30%); and (e) precipitation decreased by 60% (−60%). Each treatment consisted of four replicate plots. To avoid mutual interference, there were buffer strips of 2 m between the plots.

### 2.3. Analytical Methods and Statistical Analysis

The species number, coverage and density of these plants in the different sample plots were examined monthly from April 20, 2016 to August 16, 2016. Five sampling plots of 50 cm × 50 cm were set in each plot (one sampling plot was placed at the center, and the four others were placed near the corners). The rainfall was 34.4 mm on July 28, after which there was no precipitation for 15 days. Leaves of *B. dasyphylla* and *T. terrestris* were randomly cut on days 1, 4, 7 and 10 after the rainfall event. Several of them were taken immediately to the laboratory to measure the relative water content (RWC) of the leaf; the rest were placed in a liquid nitrogen tank for an observation of enzyme activities and osmoregulatory substances.

Leaf samples were extracted with a chilled buffer (50 mM phosphate, 1% (*w*/*v*) polyvinylpolypyrrolidone) and were centrifuged at 15,000 *g* for 20 min. The supernatants were stored at 4 °C for physiological indices. The malondialdehyde (MDA) content was measured using the thiobarbituric acid method [26]. Peroxidase (POD) activity was determined spectrophotometrically at 470 nm due to the oxidation of guaiacol. Superoxide dismutase (SOD) activity was measured spectrophotometrically using the nitroblue tetrazolium photoreduction method. Catalase (CAT) activity was determined with the iodine–perhydrol method [24,27,28]. The soluble proteins content was measured spectrophotometrically using the Coomassie blue dye combination method. The soluble sugars content was determined with the method described by An et al. [29]. The free proline concentration was measured spectrophotometrically with the ninhydrin method [30,31]. The activities of POD and SOD, as well as the amount of MDA and osmoregulatory substances were determined using a spectrophotometer (Shimadzu Corporation, Japan).

All variables (density, RWC, MDA, antioxidant enzymes and osmotic regulatory substances) were analyzed and assessed using SPSS version 20.0 and Analysis of Variance (ANOVA). A least significant difference (LSD) test was performed to determine any differences in drought resistance between the two species. Significant differences were defined at *P* < 0.05.

## 3. Results

### 3.1. Characteristics of Rainfall and Plant Growth

Rainfall patterns have an important influence on plant coverage, species composition and plant population size in the study area. The annual rainfall measured at the Naiman Desertification Research Station in 2016 was 392.8 mm, 69.3% of that rainfall falling from June to August (summer), with 84.2 mm, 128.0 mm and 60.2 mm falling in June, July and August, respectively (Figure 2A). The average coverage of plants increased from the −60% treatment to the +60% treatment (Figure 2B). The coverage was found to be only 5% in treatment −60% in June, and it reached the most (58%) under treatment +60% in July. The mean plant coverage in July for all treatments was the highest compared to that in June and August, with significant differences among the control, −60% and +60% treatments. In June, the coverage increased by 60% in treatment +30% and decreased by 40% in treatment −30% compared with that in the control. In August, the coverage was 53% in treatment +30% and 51% in treatment +60%, with no significant difference between the two precipitation treatments, but they were found to be significantly higher than those in the other treatments.

The rainfall in July was much higher than that in June and August, but the plant density in June was significantly higher than in July and August (Table 1). All four species, *T. terrestris*, *B. dasyphylla, S. collina* and *C. glaucum* were found in each of the plots. However, the dominant species under these precipitation addition treatments was *B. dasyphylla*, accounting for 44.9% of the plants under treatment +60% in August. *T. terrestris* became dominant under treatment −60% and ranged up to 80.0% in July and 79.4% in August.

### 3.2. Changes in Leaf RWC and MDA

The relative water content (RWC) and MDA are widely used to assess cellular damage. RWC is a relatively sensitive index of leaf water status, and the rate of decline under drought stress is related to drought resistance. When plants suffer from extremely high temperature, drought, salt and high radiation stress, the balance of the active oxygen metabolism system is disrupted, leading to the accumulation of reactive oxygen species (ROS). ROS accumulation causes metabolic disorder and cell membrane lipid peroxidation. MDA is the first product in cell membrane lipid peroxidation, and is widely used to determine the extent of this cell damage [32,33,34]. RWC in the leaves of both species decreases with time. The RWC of *T. terrestris* is higher than that of *B. dasyphylla* and decreases by 18.3–30.3% in the drier treatments (Figure 3A,B), while the RWC in *B. dasyphylla* decreases by 26.2–47.6%. The RWC of both species shows three phases: A slow decline (days 1 to 4), followed by a rapid decrease (days 4 to 7), and finally a moderate drop (days 7 to 10). RWC in all treatments on day 1 does not differ significantly. On day 10, *B. dasyphylla* has the lowest RWC (43.95%) under the −60% treatment, whereas RWC in *T. terrestris* is seen at 62.20%.

There is no significant difference in MDA content between the studied species on day 1, while MDA content in the precipitation reduction treatment is higher than that in the precipitation addition treatment from days 4 to 10 (Figure 3C,D). The MDA content of *T. terrestris* increases continuously over the whole stressed period except for the precipitation addition treatment, but is lower than that of *B. dasyphylla*. The MDA content in the control, precipitation reduction and precipitation addition treatments differs significantly from days 4 to 7. Under the −60% treatment, the MDA content was 2.927 (day 7) and 3.307 mmol g^−1^ FW (day 10). The MDA content of *B. dasyphylla* increases from days 1 to 7 and then decreases. On day 7, the content in treatment −60% becomes the highest (4.565 mmol g^−1^ FW), being 2.74 times that of the control and 1.90 times that of the +60% treatment.

### 3.3. Changes in Antioxidant Enzymes

POD activity in *T. terrestris* and *B. dasyphylla* was weak, with values below 1 U g^−1^ FW (Figure 4A,B), but POD activity in *T. terrestris* was relatively higher than that in *B. dasyphylla*. The activity of *T. terrestris* first increased (days 1 to 4), and then decreased (days 4 to 7), and finally increased again (days 7 to 10), except for in treatment +60%. On day 10, POD activity in treatment −60% was found to be the highest (0.709 U g^−1^ FW) and was significantly different from the control and precipitation addition treatments. The POD activity of *B. dasyphylla* first decreased (days 1 to 4), then increased (days 4 to 7), and finally decreased again (days 7 to 10) except for in treatment +60%. No significant difference in POD activity was detected between the control and +60% treatment.

The SOD activity of *T. terrestris* was significantly higher than that of *B. dasyphylla* (Figure 4C,D). The SOD activity of *T. terrestris* first increased (days 1 to 4), then decreased (days 4 to 7), and finally increased (days 7 to 10). From days 4 to 10, SOD activity in treatment −60% was higher than SOD activity in the other treatments. SOD activity in the precipitation addition treatment was significantly lower than that under precipitation reduction from days 1 to 7. The SOD activity of *B. dasyphylla* first decreased (days 1 to 4) and then increased (days 4 to 10). The SOD activity in treatment −60% was the highest (155.041 U g^−1^ FW) on day 10 and was higher than in other treatments. The SOD activity in the precipitation reduction treatment was significantly higher than that in the precipitation addition treatment from days 7 to 10.

From days 0 to 4, CAT activity of *T. terrestris* was significantly higher than that of *B. dasyphylla* (Figure 4E,F). The activity of *T. terrestris* first increased and then decreased, except for in treatment +60%. On day 10, CAT activity in treatment −60% was the lowest (37.418 U g^−1^ FW) and was significantly lower than that in the other treatments; CAT activity under precipitation addition was significantly higher than that in precipitation reduction. The CAT activity of *B. dasyphylla* in precipitation reduction first decreased (days 0 to 4) and then increased (days 4 to 10), but the activity in treatment +60% increased during the whole study period. On day 10, CAT activity under precipitation addition and precipitation reduction was significantly higher than in the control, while the activity in treatment +60% was found to be the highest (108.782 U g^−1^ FW), with significant differences from all other treatments.

### 3.4. Changes in Osmoregulatory Substances

The soluble proteins content of *T. terrestris* was significantly higher than that of *B. dasyphylla* (Figure 5A,B). From days 0 to 10, the content of *T. terrestris* was 23.212–40.733 mg g^−1^ FW versus 9.541–23.171 mg g^−1^ FW in *B. dasyphylla*. The soluble proteins content of *T. terrestris* first increased (days 0 to 7) and then decreased (days 7 to 10). On day 7, its content in treatment −60% (40.733 mg g^−1^ FW) was significantly higher than that in the other treatments. The soluble proteins content in the precipitation reduction treatment of *B. dasyphylla* increased, and was significantly higher than that in the precipitation addition from days 4 to 10. On day 10, the soluble proteins content in treatment −60% (23.171 mg g^−1^ FW) was the highest. From days 0 to 10, the soluble proteins content in treatment +60% continuously increased from 11.406 to 12.262 mg g^−1^ FW, but the differences were not significant.

The soluble sugars content on day 4 of *T. terrestris* and *B. dasyphylla* in the control was higher than those in the precipitation addition and reduction treatments (Figure 5C,D). The soluble sugars content of *T. terrestris* increased continuously. From days 4 to 10, the soluble sugars content in the control ranged from 25.556 to 27.986 µg g^−1^ FW and the differences were not significant, but were significantly higher than on day 1. On day 10, the soluble sugars content under precipitation addition was significantly lower than that in the precipitation reduction treatment and higher than that in the control; its content (48.145 µg g^−1^ FW) in treatment −60% was the highest, and was significantly different from all other treatments. The soluble sugars content of *B. dasyphylla* continued to increase in the control and +30% treatment. Its soluble sugars content first increased (days 0 to 7) and then decreased (days 7 to 10) in the precipitation reduction and +60% treatments. The soluble sugars content of *B. dasyphylla* in the control was lower than that in the precipitation reduction treatment, but higher than that in precipitation addition treatments. On day 7, the soluble sugars content in treatment −60% was 53.945 µg g^−1^ FW and was significantly different from the other treatments.

The free proline content of *T. terrestris* was significantly higher than that of *B. dasyphylla* (Figure 5E,F). Under −60% treatment, the free proline content of both species increased, but the content of *B. dasyphylla* was significantly lower than that of *T. terrestris.* From days 4 to 7, the free proline content of *T. terrestris* in the precipitation reduction treatment continued to increase and showed significant differences from the other treatments. The proline content in treatment −60% was 6.35–18.76 times and 4.50–10.63 times higher than in other treatments on days 7 and 10, respectively. From days 4 to 10, the free proline content of *B. dasyphylla* in treatment +60% did not differ significantly from the control, but was significantly higher than its free proline content on day 1. Under treatment −60%, the free proline content on day 10 was 5.54 times higher than on day 7 and was 6.27–20.99 times higher than those in the other treatments.

### 3.5. Correlation Analysis

The results of the correlation analysis (Table 2) show that RWC was significantly positively correlated CAT activity of *T. terrestris*, but negatively correlated with the other parameters in both studied psammophytes. For *T. terrestris*, MDA was significantly positively correlated with the three osmotic regulators, but negatively correlated with CAT. POD was significantly positively correlated with soluble proteins and SOD, but CAT was significantly negatively correlated with soluble sugars and prolines. Proline was significantly positively correlated with SOD, soluble sugars and proteins. For *B. dasyphylla*, there were significant positives correlations among the three osmotic regulators and they were significantly positively correlated with POD, SOD and MDA. CAT was significantly positively correlated with soluble sugars and SOD, and MDA was significantly positively correlated with the three antioxidase activities.

## 4. Discussion

Precipitation is one of the most important ecological factors that shapes plant community succession [35]. Precipitation patterns and quantities are decisive factors in plant colonization and growth. Changes in precipitation have different effects on sand area plant community structure, biodiversity and productivity, and therefore determine the characteristics of the whole ecosystem [36]. It is shown that in spare vegetation coverage areas where precipitation is less than 100 mm per year, plant coverage increases with the increase in precipitation [34]. In our study area, the south-eastern part of Horqin Sandy Land, eastern Inner Mongolia, precipitation also significantly affected the rate of vegetation cover and the vegetation density. In this area, the beginning of vegetation growth is in May, while most precipitation falls in June, July and August (summer). The precipitation in July accounts for one-third of the annual rainfall. After plants germinate in May, later precipitation significantly affects the growth of the vegetation, which is presented in vegetation coverage and density.

It is shown in this study that in all precipitation manipulation treatments, the average vegetation coverage reaches its maximum in July. In addition, an increase along the gradient from precipitation reduction throughout the control to precipitation addition treatment is found. This result is consistent with Meserve’s study, which explored the relationship between the coverage of desert ephemeral plants and precipitation in northern Chile for 13 years inclusive (1989–2001) [37]. Our study shows that coverage and other indices of the plants in August decreases in the +60% treatment and are higher than those in the control, but are lower than those in the +30% treatment. As most plants in our study area were psammophytes, suitable precipitation increases vegetation coverage and density, but high-intensity precipitation inhibits plant growth. A high precipitation rate causes these psammophytes to reduce their coverage and density, this phenomenon was also observed for other vegetation characteristics. *T. terrestris*, *B. dasyphylla, S. collina* and *C. glaucum* were present in all the plots, with significant differences in their density. The reason for this difference might be their different responses to precipitation during seed germination and during the growing season [38]. *T. terrestris* is in the Zygophyllaceae family, and the other three plants belong to Chenopodiaceae [39,40,41], but are of different genera.

In the early stages of drought, RWC and MDA were not significantly different among the different treatments. With increasing drought stress, RWC decreased; RWC with precipitation addition was found to be higher than that under precipitation reduction. Precipitation addition led to an increase in RWC, but this response lagged behind the precipitation by 1–4 days. On day 10, RWC in *B. dasyphylla* in treatment −60% was significantly lower than that found in *T. terrestris*. Fresh leaves had the ability to store and retain water, which was one of the reasons for the high survival rate of *T. terrestris* under drought stress. Under extreme drought (−60%), The MDA of *T. terrestris* reached its highest amount, indicating that *T. terrestris* suffered the most severely. The MDA of *T. terrestris* continued to increase over the whole stress period except for precipitation addition, but MDA in *B. dasyphylla* was higher than that in *T. terrestris*. This finding leads to the idea that *T. terrestris* has a better water retention capacity and a lower membrane lipid peroxidation than does *B. dasyphylla*, which makes *T. terrestris* more adaptable under drought stress. This coincides with the result that the dominant species found in treatment −60% was *T. terrestris.* It is shown that precipitation addition protects the plants from cell injury. With prolonged drought stress, the MDA of *B. dasyphylla* decreased from days 7 to 10, likely because of the extreme drought that seriously damaged its cells beyond the range of drought stress that *B. dasyphylla* could withstand.

The balance of active oxygen metabolism in plants is destroyed when plants are subjected to environmental stress [42]. However, the accumulation of oxygen free radicals can induce the antioxidant enzyme system. The synthesis of antioxidant enzymes in cells can eliminate or alleviate the toxic effects of reactive oxygen species and thus relieve the damage to the organism [43]. 

At the same time, the plants also reduce cytoplasmic infiltration by increasing the solute content of the cytoplasm to prevent cytoplasmic leakage, thus maintaining cell swelling and growth [44]. The three antioxidant enzyme activities of *T. terrestris* were stimulated and triggered at the early stage of drought. CAT activity was decreased except for treatment +60% on day 10, implying that the resistance of *T. terrestris* to drought stress via CAT activity was restricted. CAT activity cannot provide good protection under extreme drought stress. The antioxidant enzyme activities in *B. dasyphylla* remained at a low level in the early period of drought. This was probably because antioxidant enzymes were not stimulated under mild drought, whereas the activities of *B. dasyphylla* in SOD and CAT increased with prolonged drought stress. In addition, it was found that the activities of POD and SOD in *T. terrestris* were the highest under severe drought. The accumulation of oxygen free radicals activated the antioxidant enzyme protection system, and the activity of antioxidant enzymes increased, which maintained the balance of oxygen free radical metabolism. The antioxidant enzyme system is commonly assessed to reveal the ability of plants to resist stress [45]. The activities of POD and SOD in *T. terrestris* were higher than those in *B. dasyphylla*, thus, the drought resistance of *T. terrestris* was stronger than that of *B. dasyphylla.* In addition to CAT in *T. terrestris*, the antioxidant enzyme activities and osmoregulation substances of the studied species were positively correlated. To prevent excessive water loss under environmental stress, plants usually maintain cell expansion via decreasing cytoplasmic osmotic potential or increasing osmoregulatory substance content [24]. In our study, we found that osmoregulation substances were negatively correlated with RWC. The osmoregulation substance contents of the studied species increased at the early stage of drought. With prolonged drought stress, the free proline content of the two psammophytes and the soluble sugars content of *T. terrestris* increased by several dozen times, but the soluble sugars content of *B. dasyphylla* was reduced under severe drought. Drought inhibited photosynthesis and the stored soluble sugars in leaves was consumed as an energy source. The main osmoregulatory substances of *T. terrestris* were soluble proteins and free proline while soluble sugars and free proline were the main osmoregulatory substances in *B. dasyphylla.* The differences in the physiological adaptation strategies of the two psammophyte species are controlled by heredity, which is the basis for the emergence of psammophytes as target genes for stress-resistance breeding.

## 5. Conclusions

According to the above analysis, it was concluded that precipitation reduction significantly reduced vegetation coverage and density. The average vegetation coverage increased with the increase of precipitation, and reached a maximum in July. The +60% precipitation treatment was conducive to early plant growth, but not to vegetation coverage or density in the later period. Our results indicate that *T. terrestris* is better adapted to drought stress than *B. dasyphylla* in Horqin Sandy Land, eastern Inner Mongolia. The MDA of *B. dasyphylla* was higher than that of *T. terrestris*. POD, SOD, soluble proteins and free proline of *T. terrestris* were higher than those of *B. dasyphylla*. The free proline content of *T. terrestris* and *B. dasyphylla* increased several folds with prolonged drought stress. The changes in RWC content in *T. terrestris* leaves were small, and the antioxidant enzymes and osmoregulatory substances increased greatly, which effectively alleviated the damage to the cell membrane. Future studies on the adversity stress of both studied plants will evolve from the eco-physiological level to the molecular level.

## Figures and Tables

**Figure 1 plants-08-00244-f001:**
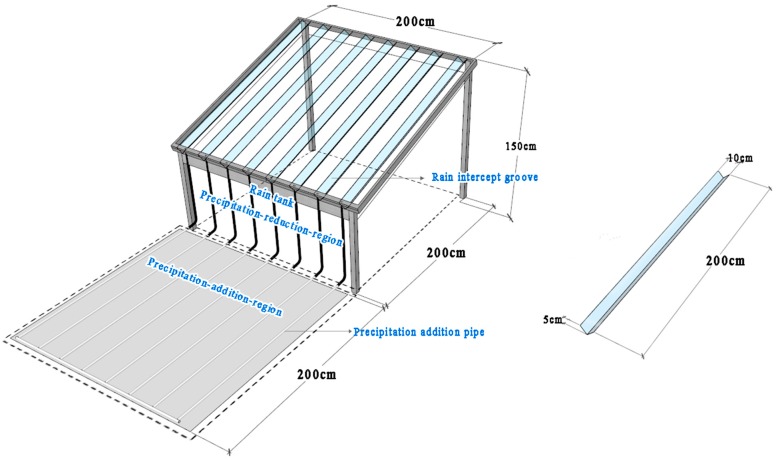
Precipitation addition-reduction apparatus.

**Figure 2 plants-08-00244-f002:**
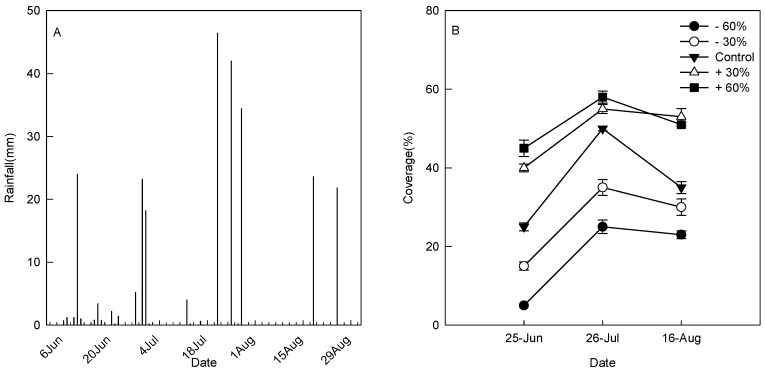
Rainfall change during the study period (**A**) and the influence of different precipitation treatments on plant coverage (**B**).

**Figure 3 plants-08-00244-f003:**
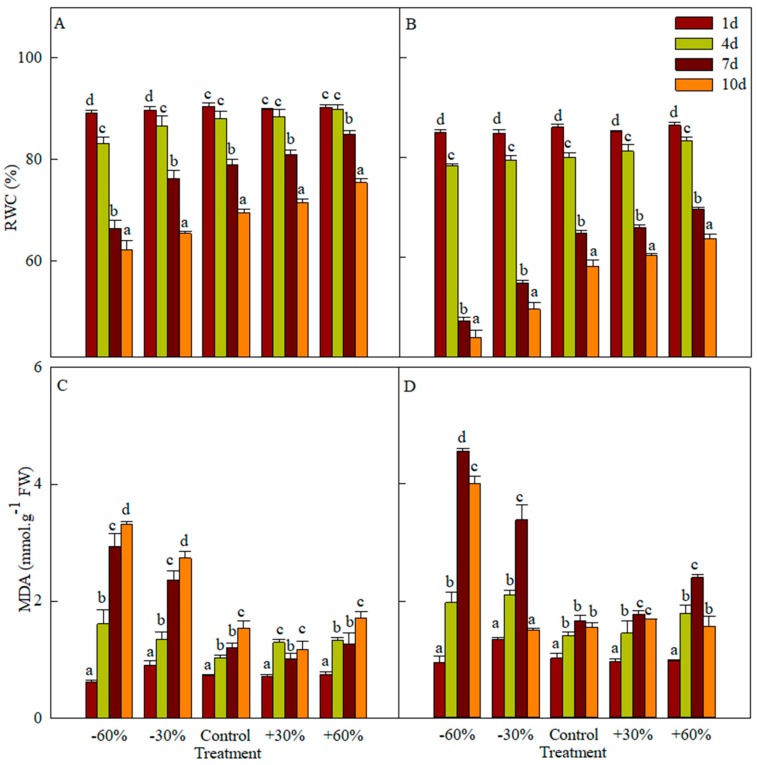
Influence of different precipitation treatments on relative water content (RWC) in *T. terrestris* (**A**), *B. dasyphylla* (**B**) and malondialdehyde (MDA) of *T. terrestris* (**C**), *B. dasyphylla* (**D**). Values are assigned as the mean ± SD.

**Figure 4 plants-08-00244-f004:**
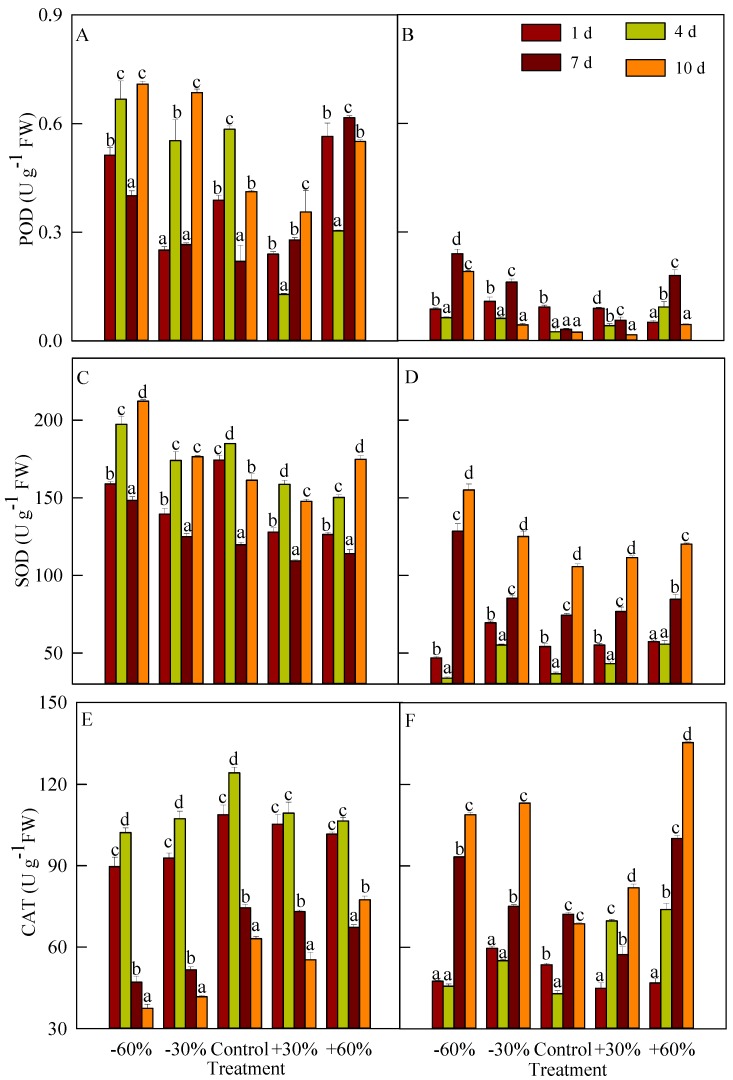
Influence of different precipitation treatments on antioxidant enzymes. (**A**): Peroxidase (POD) of *T. terrestris*; (**B**): POD of *B. dasyphylla*; (**C**): Superoxide dismutase (SOD) of *T. terrestris*; (**D**): SOD of *B. dasyphylla*; (**E**): Catalase (CAT) of *T. terrestris*; (**F**): CAT of *B. dasyphylla*. Values are assigned as the mean ± SD.

**Figure 5 plants-08-00244-f005:**
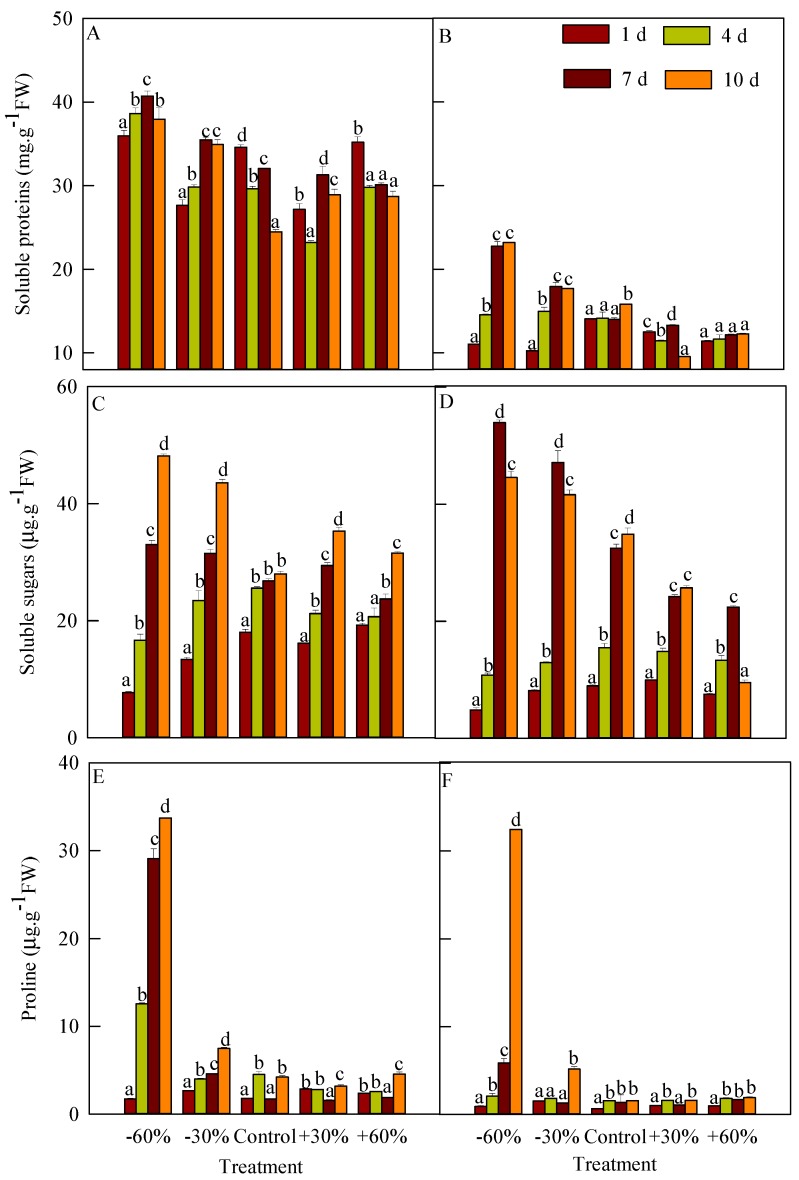
Influence of different precipitation treatments on osmoregulatory substances. (**A**): Soluble proteins of *T. terrestris*; (**B**): Soluble proteins of *B. dasyphylla*; (**C**): Soluble sugars of *T. terrestris*; (**D**): Soluble sugars of *B. dasyphylla*; (**E**): Free proline of *T. terrestris*; (**F**): Free proline of *B. dasyphylla*. Values are assigned as the mean ± SD.

**Table 1 plants-08-00244-t001:** Influence of different precipitation treatments on plant density.

Time	Treatment	*T. terrestris*		*B. dasyphylla*		*S. collina*		*C. glaucum*		Other Species		Total
		Individuals	Ratio (%)	Individuals	Ratio (%)	Individuals	Ratio (%)	Individuals	Ratio (%)	Individuals	Ratio (%)	
25-June	−60%	217 ± 3.06 D	70.0	30 ± 1.73 B	9.7	52 ± 2.65 C	16.8	5 ± 3.79 A	1.5	6 ± 3.00 A	1.9	309
	−30%	185 ± 1.53 D	46.8	84 ± 8.14 B	21.4	107 ± 3.06 C	27.2	11 ± 3.79 A	2.7	7 ± 4.16 A	1.9	394
	Control	159 ± 2.00 E	34.6	151 ± 6.24 D	32.8	130 ± 5.03 C	28.4	15 ± 4.36 B	3.3	4 ± 2.08 A	0.9	460
	+30%	136 ± 3.61 B	24.4	180 ± 4.51 C	32.4	213 ± 3.61 D	38.5	17 ± 6.00 A	3.1	9 ± 6.00 A	1.6	556
	+60%	104 ± 2.65 B	19.3	157 ± 3.61 C	29.2	238 ± 5.29 D	44.2	21 ± 6.24 A	3.9	18 ± 3.00 A	3.3	538
26-July	−60%	186 ± 3.21 D	80.0	16 ± 3.61 B	6.9	23 ± 2.08 C	10.0	2 ± 1.53 A	1.0	5 ± 2.65 A	2.1	233
	−30%	138 ± 5.69 D	52.5	69 ± 2.31 C	26.3	35 ± 2.65 B	13.3	7 ± 2.08 A	2.5	14 ± 8.74 A	5.4	264
	Control	96 ± 3.06 D	36.1	72 ± 2.65 B	27.0	80 ± 4.04 C	30.1	11 ± 2.65 A	4.1	7 ± 1.73 A	2.6	267
	+30%	76 ± 3.21 B	23.5	114 ± 4.04 D	35.4	87 ± 3.51 C	27.2	24 ± 7.00 A	7.5	21 ± 5.86 A	6.4	321
	+60%	87 ± 2.65 C	22.1	148 ± 2.08 E	37.6	113 ± 3.79 D	28.8	16 ± 6.11 A	4.1	29 ± 6.66 B	7.4	394
16-August	−60%	144 ± 3.06 E	79.4	12 ± 1.15 C	6.4	18 ± 3.06 D	10.1	1 ± 0.00 A	0.6	6 ± 1.15 B	3.5	182
	−30%	108 ± 2.08 E	46.9	67 ± 2.65 D	29.0	42 ± 2.65 C	18.2	2 ± 2.31 A	1.0	11 ± 3.21 B	4.9	231
	Control	58 ± 7.00 B	25.9	79 ± 1.53 D	35.1	69 ± 1.53 C	31.0	7 ± 1.15 A	3.0	11 ± 3.51 A	5.1	224
	+30%	26 ± 2.08 A	10.5	82 ± 6.08 B	33.7	91 ± 2.08 B	37.3	25 ± 7.00 A	10.3	20 ± 5.57 A	8.2	243
	+60%	17 ± 2.08 A	6.2	120 ± 4.58 C	44.9	77 ± 1.73 B	28.8	29 ± 6.51 A	10.7	25 ± 6.66 A	9.2	267

Values represent the mean ± SD. Values with different letters within rows are significantly different at *P* < 0.05. The ratio is the number of plants of one species to the total number of plants.

**Table 2 plants-08-00244-t002:** Pearson correlation matrix of physiological characteristics of *T. terrestris* and *B. dasyphylla* under precipitation manipulation.

Species	Item	RWC	POD	SOD	CAT	MDA	Proteins	Sugars
*T. terrestris*	POD	−0.300						
	SOD	−0.276	0.590 **					
	CAT	0.902 **	−0.180	0.053				
	MDA	−0.833 **	0.352	0.394	−0.743 **			
	Proteins	−0.320	0.477 *	0.225	−0.353	0.483 *		
	Sugars	−0.870 **	0.301	0.256	−0.769 **	0.808 **	0.213	
	Proline	−0.642 **	0.386	0.489 *	−0.512 *	0.810 **	0.603 **	0.559 *
*B. dasyphylla*	POD	−0.356						
	SOD	−0.276	0.590 **					
	CAT	−0.713 **	0.286	0.826 **				
	MDA	−0.720 **	0.787 **	0.593 **	0.448 *			
	Proteins	−0.736 **	0.594 **	0.596 **	0.366	0.815 **		
	Sugars	−0.917 **	0.468 *	0.731 **	0.497 *	0.779 **	0.803 **	
	Proline	−0.537 *	0.458 *	0.606 **	0.418	0.588 **	0.666 **	0.454 *

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
