# Peer review of "Growth and Physiology of Two Psammophytes to Precipitation Manipulation in Horqin Sandy Land, Eastern China"

_plants, 2019, doi:10.3390/plants8070244_

Reviewer 1 Report

This seems to be a well-conceived, conducted, and reported study.  While the results are not surprising, they are reasonably clear, and may be of use to readers working in similar environments.

This manuscript conducted a simple, but effective experiment of adding and subtracting precipitation in replicated plots to compare the tolerance of dry conditions between two plant species, and also measured some anti-oxidant and osmoregulatory compounds and relative water content to identify some physiological attributes leading to increase tolerance of dry conditions.  The results indicated clear patterns of differences between the species.  The experiment was clearly described and reported, and produced useful new information.

Author Response

Dear reviewer:

Thank you for your comments concerning our manuscript entitled “Growth and physiology of two psammophytes to precipitation manipulation in Horqin Sandy Land, eastern China”. We appreciate your good comments and hard work.

Once again, Thank you very much for your great efforts.

Best regards

Reviewer 2 Report

Dear Authors,

The manuscript contains interesting information on relation of plants growth and precipitation in sandy areas. In general, the study is designed and conducted well. The data are well analyzed and presented. However, there are some points which can be improved or corrected. Please consider the following points and comments in order to improve your manuscript.

Introduction: it is missing a section in the introduction with some botanical/ecological information of the target plants (life form, habitat, size of the plant, …). At the end of this section you need to shortly explain why you have chosen these two plants.

Line 42: please use simply ‘Sand ecosystem’ instead of Sandy land ecosystem’. ‘Sandy land’ is not an exact, scientific term. Please replace it in the whole text with one of the following suggestions: sand habitat, sand dune, sand area

Line 42: start the new sentences from ‘plant growth’

Line 90: what is ‘land’ habitat? Do you mean ‘sand’ habitat?

L121-123: it should be removed from the study design as it belongs to results.

L124-126: Why these two plants?

L155: study area instead of sandy land

L169: Table 1: please use a line to separate the different days (between 25-Jun and 26-Jul, and so on). It eases to read the five better to write treatments on each date.

Table 1: it is better to write number of individuals or only individuals instead of ‘Number’ in the headline of the table.

L181: Fig. 3A & B or Fig. 3A,3B

L284: remove ‘and’. Start a new sentence from ‘precipitation’.

L307: why did you not mention the family of other three species? Based on new classification, they belong to Amaranthaceae or Chenopodiaceae based on the old classification. Please add the family and mention which nomenclature reference you have used for the plants name and family.

L304-307: you have argued that differences in density can be due to their differences in response to environmental conditions as they belong to different family of plants. So, then this question comes up that why did you compare two plants which belong to different families? How do you explain that?

L309-315: I believe this explanation fit better in result part, where you present the results of RWC and MDA. It could be a good introductory for that section in order to better understand the results.

Author Response

Dear reviewer:

Thank you for your comments concerning our manuscript entitled “Effects of drought and rehydration on the chlorophyll fluorescence parameters and physiological responses of Artemisia halodendron”. Those comments are very helpful for revising and improving our paper, as well as the important guiding significance to our researches. We have studied comments carefully and made correction. The main corrections in the paper and the responds to the reviewers¢ comments are as follows.

The manuscript contains interesting information on relation of plants growth and precipitation in sandy areas. In general, the study is designed and conducted well. The data are well analyzed and presented. However, there are some points which can be improved or corrected. Please consider the following points and comments in order to improve your manuscript.

Introduction: it is missing a section in the introduction with some botanical/ecological information of the target plants (life form, habitat, size of the plant, …). At the end of this section you need to shortly explain why you have chosen these two plants.

As reviewer suggested that we have added the missing section and the reasons in the manuscript.

Tribulus terrestris is distributed in sandy land, uncultivated land, hillsides, and residential areas. There are also many distributions in agricultural and forestry production land in the protected areas and the interlaced zone between the oasis edge and the agricultural area. Its stems lie flat and up to about 1m in length [19]. B. dasyphylla is a pioneer plant in semi-fixed or fixed dunes, flat sandy land and moderately saline-alkali land. Its stem is erect and its height is about 30-50 cm. It is often scattered or clustered in grasslands, semi-deserts and desert areas of northern China [20,21]. Both plants are annual herbs [19-21], but their coverage and density varied greatly under different precipitation, thus it is necessary to reveal the differences in their physiological levels.

Line 42: please use simply ‘Sand ecosystem’ instead of Sandy land ecosystem’. ‘Sandy land’ is not an exact, scientific term. Please replace it in the whole text with one of the following suggestions: sand habitat, sand dune, sand area

We have replaced “sandy land ecosystem” with “sand ecosystem” and replaced “sandy land” with “sand area”.

Line 42: start the new sentences from ‘plant growth’

Considering the reviewer’s suggestion, we have started a new sentence from ‘plant growth’.

Line 90: what is ‘land’ habitat? Do you mean ‘sand’ habitat?

We are very sorry for our negligence, and we have replaced “land habitat” with “sand habitat”.

L121-123: it should be removed from the study design as it belongs to results.

We are very sorry for our incorrect writing and they should be placed in the results. Because the results have contained these contents, so we have deleted them.

L124-126: Why these two plants?

These sentences are not suitable for placement in the study design section, and we have deleted them. In fact, we have determined the physiological responses of these four plants to precipitation manipulation. Considering that B. dasyphylla, S. collina and C. glaucum belong to the same family, and there was significant difference between T. terrestris and B. dasyphylla in growth and physiology, thus we selected these two plants as the object of our physiological research.

L155: study area instead of sandy land

We have replaced “sandy land” with “study area”.

L169: Table 1: please use a line to separate the different days (between 25-Jun and 26-Jul, and so on). It eases to read the five better to write treatments on each date.

According to the reviewer’s suggestion, we have added lines to separate the different days (among 25-Jun, 26-Jul, and 16-Aug).

Table 1: it is better to write number of individuals or only individuals instead of ‘Number’ in the headline of the table.

We have replaced “number” with “individuals”.

L181: Fig. 3A & B or Fig. 3A,3B

We have replaced “Fig. 3A,3B” with “Fig. 3A; Fig. 3B”.

L284: remove ‘and’. Start a new sentence from ‘precipitation’.

Considering the reviewer’s suggestion, we have started a new sentence from “precipitation”.

L307: why did you not mention the family of other three species? Based on new classification, they belong to Amaranthaceae or Chenopodiaceae based on the old classification. Please add the family and mention which nomenclature reference you have used for the plants name and family.

By consulting relevant literature, we have found that the other three plants belong to Chenopodiaceae and have added references in the manuscript.

40.  Huang, Y.X.; Zhao, X.Y.; Zhou, D.W.; Zhao, H.L.; Zhang, H.X.; Zuo, X.A.; Mao, W. Allometry of Salsola collina in response to soil nutrients, water supply and population density. Nordic Journal of Botany 2010, 27, 539-547.

41.  Tobe, K.; Zhang, L.; Omasa, K. Seed germination and seedling emergence of three annuals growing on desert sand dunes in China. Annals of Botany 2005, 95, 649-659.

42.  Liu, Y.X.; Lan, X.X.; Cao, J.; Zhang, J.H.; Lan, H.Y. Screening of qRT-PCR reference genes for Chenopodium album and C. glaucum of Chenopodiaceae. Guihaia 2016, 36, 1511-1518.(in Chinese)

L304-307: you have argued that differences in density can be due to their differences in response to environmental conditions as they belong to different family of plants. So, then this question comes up that why did you compare two plants which belong to different families? How do you explain that?

All four species were found in each of the plots. T. terrestris is in the Zygophyllaceae family, but B. dasyphylla, S. collina and C. glaucum belong to Chenopodiaceae family. Under the -60% precipitation treatment, T.terrestris accounted a large proportion of the area, but B. dasyphylla was the dominant species in the +60% treatment, so we chose these two representative plants of different families to compare. Future research will be carried out from the perspective of molecular biology to analyse differences.

L309-315: I believe this explanation fit better in result part, where you present the results of RWC and MDA. It could be a good introductory for that section in order to better understand the results.

As reviewer suggested that we have remove the explanation to the discussion.

We tried our best to improve the manuscript and made some changes in the manuscript. These changes will not influence the content and framework of the paper. And here we did not list the changes but marked in red in revised paper.

Once again, thank you very much for your comments and suggestions.